# Adolescent self-consent for vaccinations: protocol for a mixed methods systematic review

Harriet Fisher, Matthew Hickman, John Macleod, Suzanne Audrey

Population Health Sciences, Bristol Medical School, University of Bristol, Bristol, UK

**Correspondence to**
Dr Harriet Fisher;
harriet.fisher@bristol.ac.uk

## ABSTRACT

**Introduction** The recent global expansion of routine adolescent vaccination programmes has the potential to protect young people against the acquisition of infectious disease and improve their health. Although in many countries the legal framework supports young people to provide consent for medical interventions if they are considered competent, written parental consent can act as a barrier to uptake as it is frequently a condition of adolescent vaccination programmes. The aim of this systematic review protocol is to document the methods which will be used to identify, appraise and synthesise the available qualitative and quantitative evidence to address: (1) whether implementation of adolescent self-consent procedures can increase vaccination uptake and (2) the barriers and facilitators to implementation of adolescent self-consent procedures.

**Methods and analysis** Comprehensive search strategy of all relevant electronic databases for both qualitative and quantitative studies using predefined inclusion and exclusion criteria. At least two authors will independently review titles and abstracts, extract data and assess the methodological quality of eligible primary studies, resolving disagreements by consensus. Quantitative studies will be reported narratively and where possible pooled in a meta-analysis using a random-effects model. The findings of qualitative primary studies will be extracted, interpreted and synthesised to identify overarching themes as well as similarities and differences within those themes.

**Ethics and dissemination** As this systematic review involves analysis of secondary data, the study does not require ethical approvals. We will use our findings to assess whether the evidence supports the hypothesis that self-consent procedures can increase coverage of adolescent vaccination programmes. We will identify barriers and facilitators to the implementation of adolescent self-consent for vaccination and make recommendations for policy makers and practitioners in relation to consent procedures within vaccination programmes for young people.

**PROSPERO registration number** CRD42017084509.

## Strengths and limitations of this study

► The mixed methods systematic review will answer complementary research questions about self-consent for adolescent vaccination programmes.
► Robust systematic review methodology will be used to identify, appraise and synthesise the relevant qualitative and quantitative literature.
► Lack of primary studies and heterogeneity of eligible studies in terms of study design, population and reporting may limit our ability to infer conclusions in relation to the research questions.

acquisition.[1 2] Provided sufficient coverage is achieved, the expansion of adolescent vaccination programmes may improve young people's health by protecting them from potentially life-threatening infectious diseases.

The introduction of new adolescent vaccination programmes is relevant to the debate about young people's capacity to provide consent to receive medical treatment. The United Nations Convention on the Rights of the Child recognises the right for all children and young people to participate in decision-making processes which involve them.[3] However, the WHO has acknowledged difficulties over consent for vaccination of adolescents because of their age and describes current practice through which countries are encouraged to adopt procedures that ensure parents have been informed and agreed to the vaccination.[4]

In most countries, the legal framework for consent requires parental or guardian permission for young people aged below 18 years.[4] However, the age of consent for medical interventions, such as vaccination programmes, is lower in some countries. In the UK, Canada and Sweden, young women are legally able to override parental decisions if they are considered mature enough to make and understand the consequences of, the decision to vaccinate. In Australia

## INTRODUCTION

In recent years, the number of routine vaccinations recommended during adolescence have increased and include vaccines that protect against tetanus, diphtheria, meningitis and human papillomavirus (HPV)

and theUSA, there are geographic variations of the age (12–17 years) that a young person can consent to be vaccinated. Despite young people being supported by the law to provide consent themselves, written parental consent is usually sought. In relation to the HPV vaccination programme, this has been shown to act as an important barrier preventing young women (usually aged 12–13 years) receiving the HPV vaccine, with implications for vaccination programme coverage.[5][6] Furthermore, it is a barrier with potential to reinforce health inequalities since lack of written parental consent may also be related to lower socioeconomic status and some ethnic groups.[5][7]

To examine the issue of self-consent for the HPV vaccine in more detail, a mixed-methods study has been funded by the National Institute for Health Research (NIHR) Research for Patient Benefit Programme (RfPB) in England. The study is examining the practicality, acceptability and impact of implementing new self-consent procedures for the schools-based HPV vaccination in two local authorities in the south-west of England.[8] There are three elements to the study: statistical analyses of routine data to assess the impact of self-consent on overall uptake levels and in relation to socioeconomic status, ethnicity and type of school; a process evaluation to examine the context, implementation and response to the new consent procedures and a systematic review of the evidence relating to self-consent for adolescent vaccines. The current protocol focuses on the systematic review which will run alongside, and inform, the other elements of the study.

An initial scoping search suggested a paucity of peer-reviewed evidence in relation to self-consent procedures for HPV vaccination programmes. Since issues relating to self-consent for the HPV vaccination are likely to be relevant for other vaccinations delivered during adolescence, we widened the scope of the systematic review to identify and collate the evidence across all adolescent vaccination programmes. We chose to restrict to vaccination programmes, rather than include studies related to healthcare in general, to ensure the findings were relevant to the programme of research described above. Therefore, the aim of this mixed-methods systematic review is to identify, appraise and synthesise the available qualitative and quantitative literature to gain understanding as to: (1) whether implementation of adolescent self-consent procedures can increase vaccination uptake and (2) the related barriers and facilitators to implementation of adolescent self-consent procedures.

## METHODS AND ANALYSIS

We are using mixed methods methodology within this systematic review to answer complementary research questions within one study. In addition to answering questions of the effectiveness of self-consent interventions at increasing uptake of adolescent vaccination programmes, the systematic review will also synthesise qualitative research comprising the views of young people

and relevant stakeholders to gain understanding of how self-consent procedures can be implemented effectively to increase uptake.[9] The findings from the qualitative and quantitative studies will be integrated to produce recommendations for future policy and practice.[9]

This review protocol was prepared using the Preferred Reporting Items for Systematic Reviews and Meta-Analyses (PRISMA) Protocol guidelines[10] (online supplementary file 1) and has been registered with the International Prospective Register of Systematic Reviews (PROSPERO) (Registration number: CRD42017084509).

### Search strategy

A comprehensive search strategy has been developed to capture all literature relevant to adolescent self-consent procedures for vaccination programmes by a reviewer (HF) experienced in undertaking systematic reviews in the proposed research field and discussed with members of the research team. The original search strategy developed for the Embase database has been adapted for each included database (see below) and comprises a combination of text words and the following medical subject headings (MeSH) indexing terms: 'child', 'adolescent', 'active immunization', 'immunization', 'immunization programs', 'mass immunization', 'revaccination', 'vaccination', 'diptheria vaccine', 'diptheria tetanus vaccine', 'diptheria pertussis tetanus', 'haemphilus influenzae type b vaccine', 'hepatitis b vaccine', 'meningcoccus vaccine', 'rubella vaccine', 'wart virus vaccine', 'papillomavirus vaccines', 'decision making', 'informed consent', 'parental consent', 'treatment refusal' (box 1). Study design filters or restrictions by setting will not be applied as the study aims to be inclusive in relation to study design and settings eligible for inclusion.

### Databases

To ensure all the relevant literature is captured, we will search the following 10 databases from inception to January 2018 and reupdated ahead of study completion to inform the wider research study as it progresses: Child Development & Adolescent Studies via EBSCOhost, Cochrane Central Register of Controlled Trials via The Cochrane Library, Cochrane Reviews via The Cochrane Library, Cumulative Index to Nursing and Allied Health Literature via EBSCOhost, Embase via Ovid, Health Technology Assessment Database, Medline via Ovid, PsycINFO via Ovid, Social Care Online via Social Care Institute for Excellence and Web of Science Core Collection: Social Sciences Citation Index and Conference Proceedings Citation Index-Science. All abstracts will be saved using Endnote X8.

### Inclusion and exclusion criteria

Quantitative studies will be eligible if vaccine uptake following implementation of self-consent procedures is reported for young people aged between 10 and 18 years.[11] Qualitative studies reporting the views and experiences of key stakeholders in relation adolescent

## Box 1    Embase search strategy

1. child/
2. adolescent/
3. ('Young people#' OR 'young person#' OR 'young offender#' OR adolescent# OR adolescence OR youth# OR minor# OR teen OR teens OR teenage OR teenaged OR teenager# OR juvenile# OR pupil# OR boy# OR girl# OR underage# OR daughter# or son# (school AND dropout#) OR (school AND 'drop out#') OR 'school aged').mp.
4. active immunization/
5. immunization/
6. immunization programs/
7. mass immunization/
8. revaccination/
9. vaccination/
10. diphtheria vaccine/
11. diphtheria tetanus vaccine/
12. diphtheria pertussis tetanus Haemophilus influenzae type b vaccine/
13. hepatitis B vaccine/
14. meningococcus vaccine/
15. rubella vaccine/
16. wart virus vaccine/
17. Papillomavirus Vaccines/
18. (cervical cancer or diptheria or diphtheria or diphteria or DtaP or DTP or Hep B or hepatitis or HPV or measles or MenC or MenACWY or meningitis or Meningococcal or Neisseria meningitidis or papillomavirus or pertus* or rubella or rubeola or td?ipv or tetanus or wart virus or whoop*).tw.
19. (policy OR programme*)
20. (immuniz* OR immunis* OR immunother* OR inoculat* OR innoculat* OR prophyla* OR revaccinat* OR vaccin*).mp.
21. Decision making/
22. Informed consent/
23. Parental consent/
24. Treatment refusal/
25. (assent* OR competen* OR decision-making OR decision making OR Gillick OR Fraser OR inform* consent OR mental capacity OR minor consent OR parent* consent OR permission* OR presume* consent OR treatment refusal OR self consent OR self-consent OR opt-out OR opt-in).mp.
26. 1 or 2 or 3
27. 4 or 5 or 6 or 7 or 8 or 9 or 10 or 11 or 12 or 13 or 14 or 15 or 16 or 17
28. 18 and 20
29. 19 and 20
30. 27 or 28 or 29
31. 21 or 22 or 23 or 24 or 25
32. 26 and 30 or 31

self-consent procedures will also be included. Studies related to consent procedures solely targeting parents of adolescents or early childhood and adult vaccination programmes will not be eligible for inclusion. Relevant stakeholders will vary with context but are likely to include young people, parents or primary caregivers, healthcare professionals, policy makers, community leaders and teachers.

We will include a range of study designs. To determine whether self-consent procedures can increase uptake of vaccination programmes, primary studies reporting parallel group randomised controlled trials, quasi-randomised trials, non-randomised controlled trials, controlled before and after studies, historically controlled studies and retrospective or prospective cohort studies that include a control group will be eligible. Qualitative studies which use interviews, focus groups, observations or open-ended questions allowing free-text responses in questionnaires will be included to explore views and behaviours related to young people's self-consent for vaccination.

Conference abstracts, reviews, editorials, opinion pieces, dissertations, letters and books will only be included if they present original data. There will be no language or country of origin restriction imposed and any relevant full text paper that is not written in English will be translated.

### Study selection

Two reviewers will independently assess the titles and abstracts against the predefined eligibility criteria. Full-text publications of all potentially relevant articles will be retrieved and examined for relevance. Any disagreements arising will be resolved by discussion. The reference lists and bibliographies from relevant studies and systematic reviews will be hand-searched for additional primary studies not retrieved by the electronic search.

We will use the reference management software EndNote X8 to remove duplicates and sort exclusions and inclusions. The search strategy and study selection process will be documented using a PRISMA flow diagram.[12]

### Data extraction

At least two reviewers will independently extract data from selected studies using structured and standardised data extraction forms used in our previous qualitative and quantitative systematic reviews. In instances where multiple publications relate to the same study, these will be reported together. The following domains will be retrieved: study characteristics (authors, publication year, country, aim, study time period, study design, location, type of setting, data collection period, data collection method, sampling strategy, analysis), participant characteristics (participant age, sample size, vaccination status of participants, socioeconomic indicators, race/ethnicity, gender and religion) and study results (uptake of vaccine, psychological outcomes, healthcare service use, incidence of vaccine preventable disease, views and behaviours related to self-consent procedures, authors' reported conflicts of interest and study funding sources). We will also record data relating to the possible harms resulting from self-consent procedures (eg, conflict with parents, healthcare professional anxiety). Where possible, authors will be contacted for missing or incomplete data. Disagreements will be resolved through discussion.

## Risk of bias and quality assessment

For eligible primary studies, quality assessment will be undertaken to illustrate potential sources of bias. As we anticipate the majority of eligible studies will be observational, studies will not automatically be excluded on the basis of 'low' quality assessment if they are considered to contribute relevant information. We propose using: the Cochrane Collaboration's handbook for the assessment of risks of bias for systematic review of randomised controlled studies and quasi-randomised intervention studies;[13] Risk Of Bias in Non-Randomised Studies of Interventions (ROBINS-I);[14] the NIH Quality Assessment Tool for Observational Cohort and Cross-Sectional Studies[15] and the Critical Appraisal Skills Programme criteria adapted for qualitative studies for evaluating qualitative research.[16] Quality assessment of primary studies will be undertaken independently by two reviewers and recorded in an excel spreadsheet. An overall assessment of 'high', 'medium', 'low' or 'unclear' will be assigned and reported.

## Data synthesis: quantitative studies

We anticipate that the primary quantitative studies will be reported narratively as preliminary searches specifically related to HPV vaccination programmes indicated a lack of published studies and the likelihood of heterogeneity in relation to study design and reported outcomes. However, if sufficiently similar studies are captured, we will consider combining individual study results through meta-analyses. To assess the heterogeneity between studies, we will use the Q-statistic and the $I^2$-statistics.[17] Evidence of heterogeneity will be classified as weak, moderate and strong for corresponding $I^2$ of 25%, 50% and 75%, respectively. If heterogeneity between studies is classified as weak, analyses will comprise adjusted ORs where available, with unadjusted ORs used if not reported. Analyses will be undertaken using the meta-analysis function[18] available in Stata V.15. We do not anticipate sufficient data being available to undertake subgroup analyses. However, if sufficient data were reported, we propose two subgroup analyses to compare impact of self-consent procedures by: (1) setting (healthcare vs school) and (2) age of participants (less than 14 years old vs 14 years and greater).

## Data synthesis: qualitative studies

The socioecological model[19] considers that behaviour is shaped by a complex interaction between factors operating at public policy, community, organisational, interpersonal and intrapersonal levels. In a previous qualitative synthesis, we have shown that young women's access to the HPV vaccine is shaped by decisions at different levels of the socioecological model.[5] During the analysis, we will use the socioecological model to provide a framework for understanding how barriers and facilitators operating at different levels of the model can provide access to, or prevent, young people self-consenting in the context of vaccination programmes.

To analyse the qualitative data, the methodology for thematic synthesis reported by Thomas and Harden,[20] assisted by the Framework method of qualitative data management,[21] will be used. These methods are suited to studies with a priori aims and objectives. The overall purpose of the synthesis will be to 'pool' the results from individual primary studies by initially separating the findings, coding and interpreting the text and then combining them through the identification of key themes across the studies as well as similarities and differences within those themes.[22] Thematic synthesis will be led by one reviewer reporting to the wider team about interpretation of the data as analysis progresses.

Familiarisation with the dataset will begin with reading the full papers. Pertinent sections of the text reported in each primary study will represent the basic units for analysis. Primary charts of the text will be constructed around key issues using the Framework Matrix within QSR NVivo11 software. For example, initial charts are likely to focus on 'barriers' and 'facilitators' to adolescent self-consent. The primary charts will be retained and revisited as required. Streamlined versions will be produced as the process of coding, summarising and synthesising the data progresses. In subsequent charts, key terms and phrases will be retained while repetition within studies and extraneous text are removed. During this process, overarching themes will be identified and differences or similarities explored within these emerging themes.

## Data synthesis: interrogation

The final stage of the analysis will aim, first, to test whether the recommendations developed from the qualitative studies have been addressed in evaluative studies retrieved for the review and, second, to examine whether interventions that match the recommendations result in higher uptake in vaccination.[9]

## Patient and public involvement

The Bristol Young People's Advisory Group comprises young people aged 10–17 years who are interested in healthcare and research. They meet regularly to help researchers with their projects and have been consulted about the design of the wider study and participant materials. They will also be invited to an event at the end of the study to consider findings and recommendations with the young people, parents, immunisation nurses and school staff involved in the study.

## ETHICS AND DISSEMINATION

We will not seek ethical approval for this study because the secondary data to be collected cannot be linked to individuals. As far as we are aware, this will be the first systematic review to collate evidence in relation to adolescent self-consent procedures for vaccination programmes. The review comprises part of a larger study. The findings of this review will inform the larger study evaluating the practicality, acceptability and impact of new self-consent

procedures for the schools-based HPV vaccination programme in the UK. Findings will also be used to make recommendations to improve self-consent procedures for young people in vaccination programmes. We anticipate the results of this study; this may be of interest to national and international stakeholders interested in improving uptake in adolescent vaccination programmes.

**Contributors** All authors were involved in the conception and design of the research. SA is the principal investigator. HF is the study manager and lead researcher. JM and MH advised on systematic review methodology. HF wrote the first draft and all authors contributed to the final version of the manuscript.

**Funding** This work is supported by the National Institute for Health Research for Patient Benefit (NIHR RfPB) programme (project number PB-PG-0416-20013). The work is also undertaken with the support of the NIHR Health Protection Research Unit in Evaluation of Interventions. The work was also undertaken with the support of The Centre for the Development and Evaluation of Complex Interventions for Public Health Improvement (DECIPHer), a UKCRC Public Health Research Centre of Excellence. Joint funding (MR/K0232331/1) from the British Heart Foundation, Cancer Research UK, Economic and Social Research Council, Medical Research Council, the Welsh Government and the Wellcome Trust, under the auspices of the UK Clinical Research Collaboration, is gratefully acknowledged.

**Disclaimer** The views and opinions expressed therein are those of the authors and do not necessarily reflect those of the NIHR RfPB Programme, the Department of Health or Public Health England.

**Competing interests** None declared.

**Patient consent** Obtained.

**Provenance and peer review** Not commissioned; externally peer reviewed.

**Data sharing statement** The data analysed during the current study are available from the corresponding author on reasonable request.

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
