## [Reviewer comments · BMJ Open]

ARTICLE DETAILS

TITLE (PROVISIONAL)	Adolescent self-consent for vaccinations: protocol for a mixed methods systematic review
AUTHORS	Batista, Harriet; Hickman, M; Macleod, John; Audrey, Suzanne

VERSION 1 – REVIEW

REVIEWER	Holly Groom Center for Health Research, Kaiser Permanente NW
REVIEW RETURNED	23-Jan-2018

GENERAL COMMENTS	The authors have put together a very nice framework to review approaches to adolescent self-consent to vaccination, and how the ability to self-consent may lead to improved vaccine uptake among adolescents. This is an important area of international research and this protocol provides a well-scoped description of how the review will ultimately help understanding the role of adolescent self-consent in vaccination uptake. The authors have referenced all the appropriate best practices and are clearly very familiar with the process involved in conducting a systematic review. As this is a protocol for a forthcoming review, my comments are related to the methods that are detailed in the manuscript. Recognizing that this review is not highly restrictive (it allows for inclusion of all observational study designs and potentially even includes abstract), it is still customary to develop an analytic framework that describes the mechanisms through which we expect the study questions to lead to an understanding of the outcomes of interest. It would be nice to have one included in this protocol, as well as the accompanying study questions that are being addressed. The authors have detailed 2 key questions for this review (1. Does implementation of self-consent increase vaccination uptake; 2. What are the related barriers and facilitators to implementation of adolescent self-consent). Some thoughts about these questions: 1. Are there other outcomes (more proximal) that would also be worth examining—in case you cannot fully measure changes in vaccine uptake? For example, increased medical visits, or other health-related services?2. While you probably cannot measure this in this review, it would be reasonable to include reductions in VPD as an outcome of interest on the analytic framework.3. Would it be worthwhile to add some sub-questions that examine differences in uptake according to population and setting characteristics? While you state in your manuscript that you do not anticipate being able to look at any data by sub-population, it is still good to state, a priori, that you imagine there are differences by study characteristics (especially if you are including studies from all
--

	countries) and that you will summarize them to your best ability. One good example would be uptake of HPV vaccine by gender (where it is being recommended for both sexes). 4. What about harms of self-consent? The potential harms are usually included as its own research question. It seems relevant in this review, as there could be harms identified in quantitative and qualitative analysis that should also be summarized. Some related questions: Search strategy: Do you have any terms for “school” or “school-based” vaccination? Inclusion/Exclusion criteria: You are choosing to include publications from all countries and in all languages. Reviews often limit to studies published in countries that are highly developed (per WHO) just to help remove differences that can be contributed by different degrees of development or health system conditions. You may have good reasons for wanting to include any possible publication, but this leads back to the importance of summarizing any included evidence according to these differences in population/geographic setting. Exclusion criteria: Are there any? It seems like only studies control groups are being included (which will likely result in the loss of quite a few studies) Abstract review and quality-rating: Are you using any software tools to help with the process of reviewing potential abstracts and then quality-rating studies? Data extraction: What software system are you using to capture extracted data? If you do add other outcomes of interest (beyond just vaccin
--	--

REVIEWER	Thomas Harder Robert Koch Institute, Germany
REVIEW RETURNED	16-Feb-2018

GENERAL COMMENTS	This is the protocol for an interesting systematic review on adolescent self-consent for vaccination and I am keen to see the results. The protocol is well-written and all necessary details are provided. I have only two comments regarding methodological issues: 1) Data extraction: The authors may consider to extract information on conflict of interest and funding source from the included studies, as e.g. the AMSTAR checklist uses this as a quality indicator. 2) Risk of bias assessment: The authors plan to use STROBE for quality/risk of bias assessment of observational studies. However, STROBE is a reporting checklist, not a risk of bias tool. I strongly encourage the authors to use risk of bias tools which fit to the respective study designs to address this issue. The authors may wish to consider inventories of such tools provided e.g. by Sanderson et al., Int. J Epidemiol. 2007; 36: 666; or by our group (Harder et al., BMC Med Res Meth 2014; 14:69).
---

VERSION 1 – AUTHOR RESPONSE

Reviewer 1.

As this is a protocol for a forthcoming review, my comments are related to the methods that are detailed in the manuscript. Recognizing that this review is not highly restrictive (it allows for inclusion of all observational study designs and potentially even includes abstract), it is still customary to develop an analytic framework that describes the mechanisms through which we expect the study questions to lead to an understanding of the outcomes of interest. It would be nice to have one included in this protocol, as well as the accompanying study questions that are being addressed.

- Thank you for your suggestion to include an analytical framework within the systematic review. As part of this wider programme of research examining the implementation of new self-consent procedures for the HPV vaccination programme, we are currently developing a detailed logic model which will be informed by the findings of this systematic review in addition to the results of the process evaluation (interviews and/or questionnaires with young women and relevant stakeholders) and routine uptake data. Therefore, at this stage, we would prefer to exclude a hypothetical analytic framework until we have collated comprehensive data to inform it.

1. Are there other outcomes (more proximal) that would also be worth examining—in case you cannot fully measure changes in vaccine uptake? For example, increased medical visits, or other health-related services?

- Lines 190-191. This systematic review is being undertaken as part of a wider programme of research to examine whether implementation of self-consent procedures can increase uptake of the HPV vaccination programme in the UK. As such, we would like to retain the focus of the main outcome for the quantitative research question. However, we will now also collate information related to psychological variables (e.g. data related to self-efficacy, theory of behaviour change) and healthcare service use as part of the review.

2. While you probably cannot measure this in this review, it would be reasonable to include reductions in VPD as an outcome of interest on the analytic framework.

- Lines 191-192. We agree with the reviewer that it is unlikely that primary studies examining self-consent procedures and vaccine uptake will also report longer-term reductions in vaccine preventable disease. However, as suggested, we will also include this potential outcome in the analytic framework.

3. Would it be worthwhile to add some sub-questions that examine differences in uptake according to population and setting characteristics? While you state in your manuscript that you do not anticipate being able to look at any data by sub-population, it is still good to state, a priori, that you imagine there are differences by study characteristics (especially if you are including studies from all countries) and that you will summarize them to your best ability. One good example would be uptake of HPV vaccine by gender (where it is being recommended for both sexes).

- Lines 223-225. We have included two potential sub-analyses which we believe would be of relevance to implementation of self-consent procedures if sufficient data was available: (1) setting (schools vs. health care setting) and (2) age of participants (below 14 years old and 14 years and above).

4. What about harms of self-consent? The potential harms are usually included as its own research question. It seems relevant in this review, as there could be harms identified in quantitative and qualitative analysis that should also be summarized.

- Lines 193-195: We agree that this is important information to collate and thank the reviewer for this suggestion. We will now collect information and summarise data relating to the potential negative consequences resulting from self-consent (e.g. conflict with parents, healthcare professional anxiety). However, we do not wish to include this as a separate research question.

Some related questions:

Search strategy: Do you have any terms for “school” or “school-based” vaccination?

- Lines 138-139: Globally, adolescent vaccination programmes are delivered in the healthcare and community settings, in addition to schools. As preliminary searches have highlighted a lack of primary studies, we have deliberately designed the search strategy to be inclusive, without restriction of vaccination programme by type of setting (e.g. school, healthcare). This is now clarified within the manuscript.

Inclusion/Exclusion criteria:

You are choosing to include publications from all countries and in all languages. Reviews often limit to studies published in countries that are highly developed (per WHO) just to help remove differences that can be contributed by different degrees of development or health system conditions. You may have good reasons for wanting to include any possible publication, but this leads back to the importance of summarizing any included evidence according to these differences in population/geographic setting.

- Lines 138-139: We would prefer to remain inclusive in relation to study settings eligible for inclusion to the study. We will however report the geographical location of the study in the descriptive tables to enable readers to identify which countries the vaccination programmes take place in.

Lines 223-225. From our previous research, we believe that setting (e.g. healthcare vs. schools) may be relevant as to whether, and how, self-consent procedures are implemented. As detailed in the comment above, we will include this as a potential area to explore by sub-group analysis if sufficient data is obtained.

Exclusion criteria: Are there any? It seems like only studies control groups are being included (which will likely result in the loss of quite a few studies)

- Lines 155-156: We have clarified that studies which related to implementation of consent procedures targeting parents of adolescents, or early childhood and adult vaccination programmes will not be included.

Abstract review and quality-rating: Are you using any software tools to help with the process of reviewing potential abstracts and then quality-rating studies?

- Line 179: We are using Endnote software to review abstracts.

Line 209: Results of quality assessment will be collated in an excel spread sheet.

Data extraction: What software system are you using to capture extracted data? If you do add other outcomes of interest (beyond just vaccine uptake), it is obviously important to add these outcomes to your data extraction form.

- Lines 184-186: Data extracted will be collated onto standardised data extraction forms.

Reviewer 2.

I have only two comments regarding methodological issues:

1) Data extraction: The authors may consider to extract information on conflict of interest and funding source from the included studies, as e.g. the AMSTAR checklist uses this as a quality indicator.

Lines 192-193: Thank you for this suggestion. We have now included this.

2) Risk of bias assessment: The authors plan to use STROBE for quality/risk of bias assessment of observational studies. However, STROBE is a reporting checklist, not a risk of bias tool. I strongly encourage the authors to use risk of bias tools which fit to the respective study designs to address this issue. The authors may wish to consider inventories of such tools provided e.g. by Sanderson et al., Int. J Epidemiol. 2007; 36: 666; or by our group (Harder et al., BMC Med Res Meth 2014; 14:69).

- Lines 204-207: We have removed the STROBE checklist as suggested. We have now revised the risk of bias tools to include Risk Of Bias in Non Randomised Studies of Interventions (ROBINS-I) and the NIH Quality Assessment Tool for Observational Cohort and Cross-Sectional Studies.

VERSION 2 – REVIEW

REVIEWER	Holly Groom Center for Health Research, Kaiser Permanente NW
REVIEW RETURNED	04-Apr-2018

GENERAL COMMENTS	The authors have made changes to the text which appear to address the comments that have been made by the reviewers. My only remaining comments/suggestions are these. 1. Consider adding 'mass vaccination' and 'vaccination campaign' to the search terms (the former is regularly used in the US, rather than mass immunization) 2. For abstract reviewed, i have previously used Abstrackr, an open-source free software developed at Brown University. It was built specifically for this process of uploading abstracts from a reference file, conducting dual screening of abstracts and reconciling difference of opinion between reviewers. Assuming you were planning to do this at the abstract phase, it might be useful to you. Maybe Endnote has a similar available function but, if not, may be worth considering.
--

VERSION 2 – AUTHOR RESPONSE

We thank the editor for providing us with the opportunity to respond to the further comments. Please find our response to each comment below.

1. Consider adding 'mass vaccination' and 'vaccination campaign' to the search terms (the former is regularly used in the US, rather than mass immunization)

Thank you for this suggestion. Since submission of the protocol paper in December 2017, searches of the databases were ran in January 2018 and over 5,000 titles and abstracts have already been independently double screened by the authors. However, in light of this comment we have rerun the search strategies in the respective databases to establish whether these search terms would be important to include. An additional 13 records were retrieved. On balance, we believe that the inclusion of the additional search terms does not warrant the duplication of the work already undertaken as it would be unlikely to result in additional studies eligible for inclusion.

2. For abstract reviewed, I have previously used Abstrackr, an open-source free software developed at Brown University. It was built specifically for this process of uploading abstracts from a reference file, conducting dual screening of abstracts and reconciling difference of opinion between reviewers. Assuming you were planning to do this at the abstract phase, it might be useful to you. Maybe Endnote has a similar available function but, if not, may be worth considering.

We are more familiar with the Endnote programme which has now been used to double screen the titles and abstracts.